# Treatment and Survival of Malignant Extracranial Germ Cell Tumours in the Paediatric Population: A Systematic Review and Meta-Analysis

**DOI:** 10.3390/cancers13143561

**Published:** 2021-07-16

**Authors:** Caroline C. C. Hulsker, Issam el Mansori, Marta Fiocco, József Zsiros, Marc H. W. Wijnen, Leendert H. J. Looijenga, Annelies M. C. Mavinkurve-Groothuis, Alida F. W. van der Steeg

**Affiliations:** 1Princess Máxima Center for Paediatric Oncology, Heidelberglaan 25, 3584CS Utrecht, The Netherlands; i.elmansori@students.uu.nl (I.e.M.); m.fiocco@math.leidenuniv.nl (M.F.); J.Zsiros@prinsesmaximacentrum.nl (J.Z.); M.H.W.Wijnen-5@prinsesmaximacentrum.nl (M.H.W.W.); L.Looijenga@prinsesmaximacentrum.nl (L.H.J.L.); A.M.C.Mavinkurve-Groothuis@prinsesmaximacentrum.nl (A.M.C.M.-G.); A.F.W.vanderSteeg@prinsesmaximacentrum.nl (A.F.W.v.d.S.); 2Mathematical Institute, Leiden University, 2333CA Leiden, The Netherlands; 3Leiden University Medical Center, Biomedical Data Science Department, Section Medical Statistics, 2333ZC Leiden, The Netherlands

**Keywords:** germ cell tumour, extracranial, malignant, survival, meta-analysis

## Abstract

**Simple Summary:**

Germ cell tumours are a heterogeneous group of neoplasms and are predominantly midline tumours occurring from birth to late adulthood. Suboptimal outcomes remain for several groups of patients, including adolescents, and patients with extragonadal tumours, high tumour markers at diagnosis or platinum-resistant disease. The aim of our systematic review was to explore survival rates internationally over the past two decades in order to better define future practice and treatment strategies and also to define specific subgroups with inferior outcomes. The results of our systematic review describe the heterogeneous nature of germ cell tumours in different anatomical locations, impacting on stage at presentation, treatment modalities used and survival data. Despite this heterogeneity, subpopulations can be defined which have an inferior survival and where future research and more individualised treatment would help to improve survival.

**Abstract:**

Objective: This systematic review and meta-analysis was performed to explore overall survival (OS) and event free survival (EFS) rates internationally over the past two decades and to define specific subgroups with inferior outcomes which may demand different treatment strategies. Methods: The search focused on malignant extracranial germ cell tumours (GCTs) in the paediatric population. The initial database search identified 12,556 articles; 32 articles were finally included in this review, comprising a total of 5095 patients. Results: The studies were heterogeneous, varying from single institution reports to large prospective trials. Older studies, describing eras where non-platinum-based chemotherapy regimens were used, showed clearly worse outcomes. Survival for stage I–II gonadal disease is excellent. On the other hand, patients with an initial alpha-fetoprotein (AFP) > 10,000 ng/mL or kU/L, age > 11 years and stage IV disease confer a survival disadvantage. For testicular disease in particular, lymphovascular invasion and certain histopathological subtypes, such as embryonal carcinoma (EC) and mixed malignant GCTs, survival is poorer. Survival data for sacrococcygeal and mediastinal GCTs show a heterogeneous distribution across studies in this review, independent of year of publication. Patients > 12 years presenting with a mediastinal GCT pose a subpopulation which fares worse than GCTs in other locations or age groups. This is independent of AFP levels, stage of disease or treatment protocol, and these patients may demand a different treatment strategy. Conclusions: This review describes the heterogeneous nature of GCTs in different anatomical locations, impacting on stage at presentation, treatment modalities used and survival data. Despite this heterogeneity, in line with the current developmental biology-based classification system, subpopulations can be defined which have an inferior EFS and OS and where future research and more individualised treatment would help to improve survival.

## 1. Introduction

Germ cell tumours (GCTs) are a heterogeneous group of neoplasms [1,2]. They are predominantly midline tumours occurring from birth to late adulthood [3]. All are believed to arise from totipotent primordial germ cells and derivatives thereof [4,5]. In childhood, ~50% are gonadal and ~50% extragonadal (~20% intracranial and ~30% extracranial), and clinical presentation depends on tumour site [1,6]. GCTs represent 3.5% of all childhood cancers occurring before 15 years of age [7]. Clinical presentation of malignant GCTs changes with age, especially with the onset of puberty, and shows sex-specific patterns, representing various subtypes of GCTs [8,9,10]. Clinical trials have shown that outcomes for paediatric GCTs are generally excellent and have improved dramatically since the introduction of platinum-based chemotherapy in the 1980′s [1,2,7]. 

Existing literature reveals excellent survival rates for stage I/II gonadal and stage III extragonadal disease [8,11,12,13,14]. However, suboptimal outcomes remain for several groups of patients, including adolescents, and patients with extragonadal tumours, high tumour markers at diagnosis or platinum-resistant disease [2,15,16]. 

Treatment should be tailored to avoid unnecessary mutilating surgery, to preserve fertility and to prevent toxic and long-term complications, while trying to achieve high rates of overall and event-free survival [17,18]. 

Since the 1980′s, several national protocols and international research collaborative efforts have taken place. Consecutive protocols for GCTs (Maligne Keimzelltumoren; MAKEI 83/86, MAKEI 89, MAKEI 96) of the German Society of Pediatric Oncology and Hematology used a multimodal approach with complete tumour resection, followed by three-agent cisplatin-containing chemotherapy regimens in fewer cycles per consecutive protocol [19,20]. The French Pediatric Oncology Society (SFOP) carried out tumeur germinale maligne (TGM) 85, 90 and 95 protocols [21,22,23]. The Italian Association of Pediatric Hematology and Oncology (AIEOP) instituted the TCG91 and TCGM-AIEOP-2004 protocols, aiming at avoiding further treatment with chemotherapy for low-risk testicular (TCG91) and ovarian (TCGM-AIEOP-2004) GCTs. [24,25,26]. The Brazilian Childhood Germ Cell Study Group ran prospective trials from 1991 to 2000 and 1999–2009 [27,28,29]. The Pediatric Oncology Group (POG) and UK Children’s Cancer Group (CCG) undertook two prospective intergroup studies on stage I–II ovarian GCTs [30,31] and stage III and IV ovarian GCTs [16,30,32], respectively. The Children’s Oncology Group (COG) ran their AGCT0132 study on stage I malignant ovarian GCT between 2003 and 2010 [11,12]. The United Kingdom Children’s Cancer Study Group (UKCCSG), UK GCII protocol, investigated the reduction of chemotherapy-induced toxicity. It became customary to adopt a non-chemotherapy watch and wait strategy for all stage I tumours [33]. More detailed information on these protocols is given in Figure 1. 

This systematic review aims to explore OS and EFS rates internationally over the past two decades in order to better define future practice and treatment strategies and, additionally, to define specific subgroups with inferior outcomes which may demand different treatment strategies. 

## 2. Results

### 2.1. Search Outcomes

The initial database searches identified 12,556 published articles. After removing duplicates and elimination by title and abstract and full text review, 32 studies were included in this systematic review. The selection process, based on the PRISMA schema, is detailed in Figure 2.

### 2.2. Overview of Studies

The included 32 studies comprised a total of 5095 patients (2639 male, 2027 female, 429 unknown) [4,5,6,7,8,9,10,11,12,13,14,15,16,17,18,19,20,21,23,24,25,26,27,28,29,30,31,32,33,34,35,36]. There were 22 prospective [4,5,6,7,8,9,10,11,12,13,14,15,16,17,18,20,21,26,27,31,32,33] and 10 retrospective studies change [32,34,35,36,37,38,39,40,41,42] A quality assessment of the included articles was performed, using the checklist of “The Strengthening the Reporting of Observational Studies in Epidemiology” (STROBE) statement to assess the individual quality of reporting of the included studies [43]. An overview of the quality assessment (including the STROBE checklist) is provided in Appendix A. First of all, prognostic factors for survival will be discussed for GCTs as a whole group. Thereafter, data will be presented according to anatomical localisation. For each anatomical localisation, a short description of selected papers will be given, followed by data on clinical presentation and age of first presentation, staging, histopathology, treatment and survival. Part of the included studies do not provide a definition of EFS. The studies that do define EFS most often define it as the time since moment of diagnosis until the moment of progression, recurrence, the occurrence of a second malignancy or death. If such an event did not occur during follow up, the last patient contact was taken into account. Survival data could be pooled for a limited number of papers per anatomical localisation. The pooled survival data can be found in Appendix A. For the ovarian location, data were pooled for 672 out of 1652 (41%) patients for OS and for 587 out of 1652 (36%) patients for EFS. For the testicular location, data were pooled for 1808 out of 2456 (74%) patients for OS and for 145 out of 2456 (6%) patients for EFS. For the mediastinal location, data were pooled for 97 out of 168 (58%) patients for OS and for 84 out of 168 (50%) patients for EFS. For the sacrococcygeal location, data were pooled for 80 out of 473 (17%) patients for OS. It was not possible to pool data for EFS for the sacrococcygeal location. 

## 3. Overall Survival Data

### 3.1. Anatomical Locations

Ten studies describe GCTs in mixed anatomical locations [26,27,28,29,32,33,37,40,44,45] and some have performed univariate and multivariate analyses on prognostic factors for survival. Gonadal localisation of GCTs is associated with significantly better survival (OS 91%, EFS 86–93%) than extragonadal disease localisation (OS 81%, EFS 56–90%) (*p* = 0.003, *p* < 0.01) [15,34]. Survival is the worst for thoracic tumours (OS 71%, EFS 69%) [19]. One study reports the combination of age > 12 years and having a thoracic tumour as being a highly significant factor determining OS [32]. Five studies report on disease stage significantly affecting survival (OS and/or EFS), either with stage I–II having a survival benefit compared to stage III–IV disease [26,28,33] or with survival declining for each stage of disease [44]. OS for stage I disease ranges from 92.6% to 100%, where this ranges from 54.8–60.4% for stage IV disease. 

### 3.2. Age

Younger age offers a significant (reported OS varying from 46–59% for age >12 and OS 74–88% for age < 12, *p* ≤ 0.01) [13,14] or non-significant [15] survival benefit for GCTs across all anatomical localisations; however, different cut-off ages are used varying from 1 [40] to 10 [26], 12 [28] and 14 [27] years, respectively. 

### 3.3. Serum Tumour Markers

Some studies report that patients with serum AFP levels higher than 10,000 ng/mL or 10,000 kU/L have a significantly lower survival than patients with serum AFP levels lower than 10,000 ng/mL or 10,000 kU/L (OS 59% versus OS 85%, *p* = 0.02) [26,33], while one study reports no relation of serum AFP levels with survival [44]. OS and EFS are significantly better when lactate dehydrogenase (LDH) values are within the normal range, compared to when LDH values are abnormal, according to two studies (OS for normal LDH values 77% versus OS 94% for abnormal LDH, *p* ≤ 0.01). One study reports a non-significant survival benefit [26]. 

### 3.4. Chemotherapy Regimens

There is a significant survival difference when non-platinum-containing regimens are compared to platinum-containing regimens [34,40]. Chemotherapy regimens are indicated for certain stages and localisations of disease, and some studies are specifically designed to look at survival benefit for certain disease locations or risk groups. General findings applying to all anatomical localisations of GCTs are reported in two studies [28,40]. Lopes et al. reports that for high-risk patients (failure of decline of tumour markers and/or lack of complete response and/or disease progression), adding two more cycles of cisplatin-etoposide confers a survival benefit, compared to adding two cycles of ifosfamide-vinblastine-bleomycin [14]. Suita et al. reports significantly better overall survival in patients receiving a cisplatin-vinblastine-bleomycin (PVB) regimen compared to other, non-platinum-containing, regimens [34]. Frazier et al. reports no significant difference in outcome among paediatric and adolescent extracranial GCT patients treated with cisplatin-etoposide-bleomycin (PEB) versus carboplatin-etoposide-bleomycin (JEB) when comparing groups by age, site, stage and tumour markers at diagnosis, in univariate and multivariate analyses [37]. Unfortunately, no clear data could be extracted on survival for specific histological subtypes. 

### 3.5. Stage

Lower stage disease, across all anatomical localisations, carries a better prognosis than higher stage disease. In older studies, this difference is more pronounced, possibly reflecting more refined treatment strategies leading to better survival, even for advanced stage disease in recent years. OS for stage IV disease is 54%, 55% and 57%, respectively, in studies published in 2002–2009 [26,27,28,40]. OS for stage IV disease is 70% in a study published in 2016 [29]. OS for stage I–II disease ranges from 75–100% [26,27,28,29,40]. Four studies report that OS and EFS are significantly worse when lung and/or liver metastases are present compared to lymph node or peritoneal metastases presence only [27,28,29,40]. When considering surgical treatment, one study states that for all stages and locations of tumour, complete tumour removal leads to significantly better OS and EFS, compared to surgery leaving residual disease or biopsy at diagnosis only [27,28]. Suita et al. [34] reports this finding for stage III and IV disease only, as does Lo Curto et al. [10].

## 4. Individual Anatomical Localisations

### 4.1. Study Characteristics

Twenty-one articles studied ovarian GCTs, including sex cord-stromal tumours (SCSTs) (*n* = 1652). Ten articles described only ovarian pathology [11,20,21,22,24,30,34,36,41,46], the largest studies reporting on 217 [46] and 131 [30] ovarian GCTs, respectively. Details on the studies of ovarian pathology and pooled survival data are given in Table 1 and Figure 3, respectively. 

### 4.2. Clinical Presentation

Median age at presentation of the patients included in the ovarian-only articles ranged from 6.8 to 13.0 years. Ovarian GCTs occurring below the age of one year were rare (1–2% of cases). The majority of patients presented with abdominal pain (43–75%), although a substantial proportion of patients presented with acute abdominal pain due to torsion, peritonitis or other causes (6–17%). A palpable mass was felt in between 24% and 57% of cases [20,36]. 

### 4.3. Staging

Thirteen of the studies reported on staging for ovarian GCTs, including SCSTs. The Fédération Internationale de Gynécologie et d’Obstétrique (FIGO) classification system was used in five [20,33,34,36,41], the TNM or a modified TNM staging system was used in three [21,23,26] and the COG system in four [11,24,30,31]. Different staging systems were used in one study, which analysed collaborated data from eight trials conducted by groups from different continents [46].

Out of 835 cases with known staging details, 280/835 (33%) had stage I disease, 497/835 (60%) had stage II/III disease and 58/835 (7%) stage IV disease.

### 4.4. Histopathology

Out of 1652 ovarian GCTs, histopathology information could be extracted for 718 (43%). There were 26 (3.6%) mature teratomas, and where possible, these were excluded from the survival analysis. There were 40 (5.5%) immature teratomas (four grade I, seven grade II, 17 grade III, 12 unknown grade). There were 173 (24%) pure yolk sac tumours (YSTs), 214 (30%) mixed malignant GCTs (56 YST and other mixed malignant elements, 42 teratoma and YST, 27 immature teratoma and YST, 39 teratoma and other mixed elements, 50 mixed malignant elements). There were 142 (20%) dysgerminomas, 13 (1.8%) EC, six (0.8%) choriocarcinomas, four (0.6%) gonadoblastomas and 99 (14%) SCSTs (78 juvenile and adult granulosa cell tumours, 17 Sertoli–Leydig cell tumours, two sclerosing stromal tumours, one sex cord-stromal with annular tubules, one SCST NOS). The pure or mixed (im)mature teratomas and YST, predominantly diagnosed before puberty, are currently considered paediatric (Type I) GCTs, while the others, i.e., diagnosed after puberty and containing one of the other histological elements (EC, choriocarinoma, dysgerminoma), are considered Type (II) and, by definition, malignant GCTs. In the paediatric age group, the YST is to be considered malignant as well [47,48]. The gonadoblastma is the precursor lesion of a malignant GCT of the dysgenetic gonad [49,50].

### 4.5. Treatment

Surgical management of ovarian GCTs varied among studies. For stage I/II disease, surgical management entails oophorectomy or salpingo-oophorectomy if feasible, across all studies. With regard to contralateral pathology and higher stage disease, in two studies, bilateral oophorectomy is performed [31,34], whereas most studies perform only a biopsy of the contralateral ovary. Two recent studies describe performing non-mutilating surgery, defined as sparing the uterus and contralateral ovary if feasible [22,23]. When complete resection was not feasible, and especially in the case of advanced disease, a biopsy was performed, followed by chemotherapy and delayed tumour resection [22]. 

All studies describe thorough exploration and multiple biopsies of peritoneal surfaces in case of suspicious findings, as well as sending ascites or peritoneal washings for cytological examination in order to stage and treat appropriately. Second look surgery is described in 15 studies for different reasons, ranging from removing residual tumour after initial biopsy to surgery upon deterioration of tumour markers during follow-up [4,5,6,7,10,11,13,14,15,16,18,23,25,32,33]. 

Chemotherapeutic regimens varied across countries and time periods. Regimes not including platinum-based chemotherapeutic agents are described in five studies [20,34,36,40,41].

### 4.6. Survival

Survival data on ovarian pathology specifically can be extracted from most studies. Across all articles, OS for ovarian GCT ranged from 68–100%. When the oldest study (OS across all stages 68%) [34] with non-platinum regimes was not taken into account, the lowest OS value was 73% [27], supporting the notion that platinum-based chemotherapy confers a survival benefit. EFS data ranged from 51–96%. From studies on ovarian GCT only, several survival differences in subgroups are reported. Survival is better in the age group < 11 years [20,24,34] and lower stages of disease [20,34,41]. Terenziani et al. report the worst survival for the combination of stage IV disease and age > 11 years [24]. Baranzelli et al. report high mortality among patients with an initial AFP > 15,000 ng/mL at time of diagnosis [21]. No differences in survival are reported for histological subgroups, including the study by Schneider et al. on SCSTs [11,20,34,41]. Two studies report no difference in survival for age, stage and initial AFP level [11,46]. In Table 1, survival data for ovarian GCTs is shown. 

## 5. Testis

### 5.1. Study Characteristics

Eighteen articles describe testicular pathology (*n* = 2456). Seven articles study testicular GCTs only [12,13,25,35,39,42,51]. Nine articles describe multiple anatomical locations, including testicular pathology [26,27,28,29,33,37,40,44,45]. The largest study on exclusively testicular GCTs is a population-based study from a Surveillance, Epidemiology and End Results (SEER) database, including a total of 13,963 patients, 12,467 being adults. For this review, only data on the 1496 adolescents aged 13–19 were extracted [35]. Details on the studies of testicular pathology and pooled survival data are given in Table 2 and Figure 4, respectively. The same subclassication in Type I and II is applicable for the testicular GCTs, as described for those of the ovary (see above). 

### 5.2. Clinical Presentation

Three studies make a clear distinction between paediatric and adolescent age groups [25,42,51]. Median age in the paediatric age group ranges from 16 months to two years. Median age in the adolescent age group ranges from 15 to 18 years. The majority of patients presented with a testicular mass (76–100%) or scrotal swelling (17–98%). One study describes that the preoperative diagnosis was assessed correctly as a malignant tumour in 79% of cases, but misdiagnosed as a hydrocele, inguinal hernia or acute testicular torsion in the remaining 21% of cases [13]. One study reporting on SCST describes gynecomastia upon presentation in 18% of cases [23]. 

### 5.3. Staging

Ten of the studies on testicular pathology reported on staging for a total of 505 cases. The COG system was used in six studies [12,13,25,31,33,39]. The TNM or modified TNM system was used in four studies [23,26,42,51]. Four studies report on stage I pathology only [12,13,23,42]. One study reports on stage II testicular pathology only [31]. Out of 505 patients with known staging details, 370/505 (73%) had stage I disease, 98/505 (20%) had stage II/III disease, and 37/505 (7%) had stage IV disease. 

### 5.4. Histopathology

Out of 2456 testicular GCTs, histopathology data could be extracted for 1967 (80%). There were 86 (4.3%) mature teratomas, and where possible, these were excluded from the survival analysis; two (0.1%) immature teratomas were of unknown grade. There were 1096 (56%) mixed malignant GCTs (1079 mixed malignant nonseminomatous GCT NOS, eight teratoma and EC or other malignant elements, five immature teratoma and YST, four mature teratoma and YST). There were 301 (15%) ECs, 285 (14%) pure YST, 23 (1.2%) choriocarcinomas, 3 (0.2%) teratocarcinomas, 156 (7%) nonseminomatous NOS, four (0.2%) seminomas (three in the adolescent age group 13–19 years, one of unknown age), 11 (0.6%) SCSTs (four granulosa cell tumours, three Sertoli cell tumours, one Leydig cell tumour, one Sertoli–Leydig cell tumour, two SCST NOS). The term teratocarcinoma is historical, predominantly referring to a combinatory constitution of teratoma and EC, representing mixed (Type II) GCTs, according to the most updated WHO classification (2016).

### 5.5. Treatment

Most studies describe strict surgical guidelines on testicular GCTs, detailing an inguinal approach, high ligation of the cord at the level of the internal inguinal ring, and radical inguinal orchiectomy (RIO). In two studies, a trans-scrotal orchiectomy is carried out in >10% of included cases [25,39]. Trans-scrotal biopsies are mentioned as being contra-indicated by one study [26] and as being a reason for secondary inguinal resection by one other study [23]. Hemiscrotectomy is indicated when scrotal tissues are involved according to one study [26] and when a previous scrotal biopsy or trans-scrotal orchiectomy was performed in two other studies [31,42]. Testis-sparing surgery of the less involved testicle in case of bilateral disease is described in one study in order to avoid castration [25]. 

All but three studies report on chemotherapy regimens. Of these studies, most report that no chemotherapy is administered to stage I testicular tumours after RIO, except in the case of tumour markers failing to normalize postoperatively or in the case of residual or recurrent disease [12,13,23,25,26,28,29,33,45]. Only one study has non-platinum containing regimens among different schemes used in their study [39]. Higher stages testicular disease are commonly treated with a JEB or PEB regimen. 

### 5.6. Survival

Survival data on testicular pathology could be extracted from fourteen studies [4,7,10,13,17,21,24,26,28,30,31,32,34,36]. In studies reporting on multiple anatomical locations, gonadal site is seen as favourable in terms of survival [27,33]. Mann et al. report an EFS of 100% for testicular GCTs [33]. Suita et al. show testicular GCTs have a significantly better survival than any other anatomical site [40].

Survival is worse for stage III/IV disease compared to lower-stage disease [39,51]. Stage I/II disease generally has an excellent outcome [12,13,42] with slightly worse survival rates reported in case of lymphovascular invasion, age > 11 years and mixed histology, specifically when elements of embryonal carcinoma are present [12,42]. EFS is 45% for stage I pathology, with mixed malignant histology, and 81% for stage I disease, with YST or YST mixed with (im)mature teratoma histology. EFS is 84% when lymphovascular invasion is absent and 62% when it is present, and the adverse effect of lymphovascular invasion was observed in both the paediatric and adolescent age groups [12]. 

Relapse rates are higher with scrotal violation, although this has no impact on overall survival for lower-stage disease [13,42]. Lymphovascular invasion is mentioned as a risk factor for worse relapse-free survival in another study [25].

Paediatric age groups (<11 years in most studies) have a better survival than adolescent (>11 years in most studies) age groups [12,42,51]. In the paediatric age group, EFS ranges from 78.6–87.2% and OS is 100% [51]. In the adolescent age group, EFS ranges from 48–78.6%, and OS is 84.4% [51]. Adolescents more commonly present with regional and distant metastatic disease and with embryonal and mixed histology, with worse survival compared to the paediatric population which presents with YST disease [35,39,42,51]. One study achieves similar OS for paediatric and adolescent age groups but worse EFS, and it may be that given the decreased EFS in adolescents, a higher burden of therapy was required to achieve a similar outcome [51]

It is difficult to comment on survival for the SCST. The only study with data on SCST in the testicular location is by Fresneau et al. [23]. This study reports on 11 stage I testicular SCST, all treated by surgery only and cured without relapse. See Table 2 for extracted data on EFS and OS for testicular GCTs.

## 6. Mediastinum

### 6.1. Study Characteristics

Six articles describe mediastinal pathology (*n* = 168) [19,26,32,33,39,40]. Two articles are exclusively on mediastinal GCTs [19,40], while four articles describe mediastinal GCTs as part of a cohort of GCTs in mixed anatomical locations [26,32,33,37]. Details on the studies of mediastinal pathology and pooled survival data are given in Table 3 and Figure 5, respectively.

### 6.2. Clinical Presentation

Three studies describe a bimodal distribution of mediastinal GCTs, with a peak of patients diagnosed in early childhood and a peak in presentation at adolescent age (26,33,40). This again represents the subclassification into the Type I and II GCTs, as mentioned both for the ovary as well as testes (see above). Male predominance is described in most studies but only quantified in one, with a clear male predominance of 79% [38]. Presenting symptoms are reported as various respiratory symptoms, such as cough or wheeze, with a duration of two weeks to two months. Very young patients are reported to present with respiratory distress, and older patients reported chest pain [38]. Two studies report on four patients having their mediastinal GCT diagnosed on antenatal ultrasound [19,38]. 

### 6.3. Staging

Stage can be extracted from two studies for a total of 32 cases [33,38]. See Table 3 for details. In the study by Schneider et al., nine out of 26 patients with malignant GCTs had metastases in the loco regional lymph nodes (*n* = 4), the lungs (*n* = 5), central nervous system (CNS) (*n* = 1), liver (*n* = 2) or bone (*n* = 2). In the study by Grabski et al., nine (47%) patients with malignant GCTs presented with distant metastasis. The most common metastatic site was the lungs, which occurred in seven of the nine patients (78%). 

### 6.4. Histopathology

Out of 168 mediastinal GCTs, histopathology data could be extracted for 58 (35%). There were 26 (45%) pure YST, 21 (36%) mixed malignant GCTs (four teratoma and YST, four YST and EC, one teratoma and YST and EC, six teratoma and one or more malignant elements, two YST & EC and choriocarcinoma, one YST and EC and seminoma, one seminoma and EC, one seminoma and choriocarcinoma, one unknown). There were three (5.2%) ECs, three (5.1%) choriocarcinomas and five (8.6%) seminomas. The distribution of the histopathologic sub-entities varied according to age. In infants, malignant GCTs with pure YST histology or mixed histology with predominant YST components were most prevalent. Mixed malignant GCTs, including choriocarcinoma and EC, as well as some YST components, are seen in children from age five onwards. Mediastinal seminomas are not reported in children younger than 10 years of age [19,33,38]. This dichotomy fits the forementioned subclassifcation into the Type I (teratomas and YST) and Type II GCTs (EC, YST, choriocarcinoma, teratoma) being predominantly pre- and post-pubertal regarding age at clinical presentation. This is also represented in the clinical behaviour description (see below).

### 6.5. Treatment

Surgical details can be extracted from the mediastinal-only articles. A total of 39–50% of cases underwent primary surgery. A total of 22–50% of cases had neoadjuvant chemotherapy with a reduction of tumour dimension of 12–75% followed by surgery. Surgical approach varied. In primary as well as post-chemotherapy surgery, similar perioperative findings and complications were described. One study described upfront surgery in 50% of its mediastinal GCTs, with 50% failing to achieve complete resection, whereas all cases which had delayed surgery after neoadjuvant treatment achieved complete tumour clearance, obviating the need for second-look surgery [19]. No attempt at surgery was described in 11–27% of cases. Reported reasons were complete regression on chemotherapy +/− radiation, death of complications of treatment or progression of tumour during treatment [19,40].

### 6.6. Survival

Survival details on mediastinal GCTs could be extracted from four studies [19,32,33,40]. OS for thoracic mediastinal GCTs ranges from 39–87%, and EFS ranges from 29–83%. Univariate analyses or multivariate regression models were performed in some studies. Patients with mediastinal GCTs and metastases at presentation are reported to have inferior outcomes, with an OS of 0% in one study [38] and an EFS of 65% compared to an EFS of 93% when no metastases were present in another study [19]. In Grabski’s study, all patients with metastatic disease at presentation died within five years of diagnosis [38]. Complete resection with negative margins and platinum-based chemotherapy are factors providing a statistically significant survival benefit. Patients with complete tumour resection with negative margins have an OS of 73% versus 11% for incomplete tumour resection with positive margins in one study [38]. Schneider et al. report that complete tumour resection is the strongest indicator of tumour control. For patients who underwent complete resection, EFS was 94% (19 of 20 patients; median follow-up, 68 months), and OS was 100% (20 of 20 patients; median follow-up, 75 months) [19].

In the same study, patients with mediastinal GCTs and younger age are reported to have better survival outcomes, with OS of 93% for age < 5 years versus 78% for age > 5 years [19]. The combination of age > 12 years and the thoracic primary site resulted in six times the risk of death compared with patients younger than 12 years with tumours at other sites, according to one study [32]. The authors suggest that this subset of patients have a biologically different disease. Prognosis did not vary between the groups of pure YST and tumours of mixed histology [19]. The study by Grabski et al. shows worst rates for OS and EFS. In this study, nine out of 19 malignant mediastinal GCTs (47%) presented with metastasised disease, as opposed to the study by Schneider et al., where only nine out of 26 (35%) presented with metastasised disease [19,38]. In Grabski’s study, there does not seem to be a survival benefit in the younger age group or the female population. Out of five patients < 3 years old, four died of disease (one female with pure YST and three males with EC, YST and YST/EC, respectively). In the older age group, nine out of 14 patients died, of which four were not from disease (one of myocardial infarct 20+ years later, one of lung cancer 20+ years later, one of relapse and acute myeloid leukaemia, one of intracranial aspergillus), and five were from disease (histopathology: one EC, two YST, one choriocarcinoma, one mixed YST/EC). Among the five survivors, four were male. Of note, four survivors had mixed-type malignant histology, and most patients who died of disease had EC, choriocarcinoma or seminoma histology. See Table 3 for extracted data on EFS and OS for mediastinal GCTs.

## 7. Sacrococcygeal

### 7.1. Study Characteristics

Nine articles (*n* = 473) describe sacrococcygeal pathology [26,27,28,29,32,33,37,40,45]. All articles are on GCTs in mixed anatomical locations, including sacrococcygeal locations. Details on the studies of sacrococcygeal pathology and pooled OS data are given in Table 4 and Figure 6, respectively. These predominantly reflect Type I GCTs [47,48]. 

### 7.2. Clinical Presentation

Two studies describe the age distribution for sacrococcygeal GCTs [26,33]. There is a unimodal age distribution, most patients presenting before four years of age, with a peak between six months and two years. 

### 7.3. Staging

The majority of sacrococcygeal GCTs present as stage III and IV disease [26,33]. See Table 4 for details. 

### 7.4. Histopathology

Out of 473 sacrococcygeal GCTs, histopathology data could be extracted for 37 (7.8%). This is because there are no studies on exclusively sacrococcygeal GCTs; they are only reported as part of large studies, including all anatomical locations, without reporting of histology for each anatomical subset. Out of 37 sacrococcygeal GCTs, there are 14 (38%) pure YSTs and 23 (62%) mixed histology (teratoma and one or more malignant elements) malignant GCTs [33].

### 7.5. Treatment

Few studies report on the surgical approach for sacrococcygeal GCTs, specifically. When reported on, it is recommended that resection is performed only if this can be done completely including removal of the coccyx, without causing significant surgical damage, otherwise a biopsy is indicated [26,33]. 

In five studies, data on chemotherapy for sacrococcygeal GCTs could not be extracted. In two studies, all patients with sacrococcygeal GCTs received chemotherapy [33,37], including patients with stage I and II disease [33]. In two other studies, the majority of patients received chemotherapy [28,29]. 

### 7.6. Survival

In five studies, data on EFS and OS are available. EFS for sacrococcygeal pathology ranges from 27–87%. OS ranges from 55–88%. In a multivariate analysis by Lo Curto et al., the site of the primary tumour being sacrococcygeal or other extragonadal site was the only variable statistically related to a worse prognosis [26]. All other studies which include sacrococcygeal GCTs have no extractable subgroup information on sacrococcygeal tumours. See Table 4 for extracted data on EFS and OS for sacrococcygeal GCTs.

## 8. Discussion

The current systematic review explores the treatment and survival of extracranial malignant GCTs over the past two decades. This is in spite of the more recent insights into the developmental origin of GCTs, reflected by the current classification [47,48]. The studies are a reflection of the world-wide approach of GCTs, which is varied and heterogeneous. As we aimed to include studies reporting on treatment and survival data of malignant extracranial GCTs in the paediatric population; this search yielded heterogeneous results, varying from single institution reports to population-based studies, intergroup studies and large prospective trials. In addition, it reflects the various variants of GCTs, both Type I and Type II. There has been an evolution of treatment strategies centred on platinum-based chemotherapy regimens since the 1980’s, but there is great heterogeneity in treatment protocols and the analysis of results. This makes pooling of survival data in this systematic review difficult. Therefore, pooling survival data was only possible for a limited number of studies per anatomical localisation, and for the sacrococcygeal localisation, only OS data could be pooled. 

This review demonstrates that older studies, describing eras where non-platinum-based chemotherapy regimens were used, show a clearly worse outcome across all stages and anatomical localisations of disease. It is well-known that, with the introduction of platinum-based chemotherapy regimens, survival rates for GCTs increased dramatically [52,53,54]. Higher stage disease is associated with a worse outcome, and in particular, metastasised disease to lung and/or liver is associated with the poorest outcome. The current review found that this difference is more pronounced in older studies which might reflect more refined treatment strategies employed in recent years [9]. Systematic national and international cooperative therapeutic protocols, utilising platinum-based combination chemotherapy integrated into a multimodal therapeutic approach, were introduced in the 1990s [17]. Better survival, also for advanced stage disease, could thus be attributed to changes in diagnostic techniques and therapies introduced by new protocols. 

This review concludes that survival for stage I–II gonadal disease is excellent; however, EFS for ovarian pathology is mediocre, ranging from 58–96%. What cannot be concluded, however, is which subpopulation of ovarian pathology drives this mediocre EFS. There is no survival difference across histopathological sub entities, including SCSTs, but our review suggests that an initially very high AFP > 10,000 kU/L or ng/mL, age > 11 years and stage IV disease confer a survival disadvantage. Current surgical management is based on exploration of the abdomen, looking for macroscopic abnormalities which should be biopsied. There may be a risk of under-staging with this procedure. Future research could look into strategies to improve detection of deposits in order to improve initial staging, especially prompted when presented with paediatric patients in the >11 age group and diagnosed with an initial AFP > 10,000 ng/mL. These relate to the Type II GCTs [47,48].

In testicular disease, this review shows that nearly three quarters of patients present with stage I disease, which carries excellent survival rates. Survival is poorer for the >11 years age group, again reflecting the Type II GCTs, when lymphovascular invasion is present, and for certain histopathological subtypes, such as embryonic carcinoma and mixed malignant GCTs. Older patients tend to present with more advanced stage disease according to this review. The very young age group with Type I GCTs present with yolk sac pathology and early-stage disease. They do not develop from germ cell neoplasia in situ (GCNIS], but from an early defective primordial germ cell [55]. The older age group typically indeed present with type II mixed or embryonic histological subtype, which may behave more aggressively, and hence present at a more advanced stage, which is reflected in survival data. However, one study reports similar OS data for paediatric and adolescent age groups but worse EFS for the latter. This would suggest a higher therapy burden to achieve a similar outcome. Thus, future research should focus on minimising the total amount of necessary therapy in this vulnerable population, given the potential long-term therapeutic side-effects, including cardiovascular disease and secondary malignancy.

Survival data for sacrococcygeal and mediastinal GCTs show a heterogeneous distribution across studies included this review, independent of year of publication. Survival for mediastinal tumours is dependent on metastatic status at presentation and completeness of surgery. In one study, OS was 0% for patients presenting with metastatic disease. This dismal outcome, combined with the fact that mediastinal tumours present in a more advanced stage, contributes to poorer outcome of mediastinal GCTs. In other studies of this review, mediastinal location is an independent factor contributing to poorer outcome. This review shows that patients > 12 years presenting with a mediastinal GCT pose a subpopulation which fares worse than GCTs in other locations or age groups. In other words, the clinical behaviour of Type II GCTs is also dependent on anatomical localisation, in which the mediastinal localisations do the worst. This is independent of AFP levels, stage of disease or treatment protocol. Histology may play a part in this. In line with the histological and age-related subclassification in Type I and II GCTs, mediastinal GCTs in younger patients (i.e., Type I) are commonly teratomas or pure YSTs displaying chromosomal deletions at 1p, 4 and 6q and gains at 1q, 3 and 20q. Mediastinal GCTs occurring at an older age (i.e., Type II) show patterns of chromosomal aberrations, such as a gain of 12p, loss of chromosome 13 and gain of chromosome 21 [56], and they are of mixed histology. Although, in this current review, there is not enough data to show a survival difference among histopathological subtypes, the better prognosis of malignant mediastinal GCTs in infants compared to that in adolescents suggests that there may be histopathologic and genetic differences, which are associated with differences in clinical behaviour and which may demand a different treatment strategy. Analysing this further, Oosterhuis and Looijenga discussed original data that mutational and copy number variations analysis of a large cohort of patients with primary mediastinal nonseminomas demonstrated a high prevalence of TP53 mutations, likely explaining why patients with primary mediastinal nonseminomas are frequently resistant to platinum-based chemotherapy [57,58], in line with previous findings [59]. With mediastinal tumours, as with other extragonadal GCTs, frequently presenting in advanced stages, they are dependent on chemotherapy for survival, unlike stage I disease. Resistance to platinum-based chemotherapy would impair this important mode of treatment and prevent cure by current standard treatment protocols. 

Sacrococcygeal GCTs show the widest range of EFS and OS. It must be noted that, during our search, the sacrococcygeal location was the only location which was not studied as a sole entity in the studies which our search yielded. Instead, sacrococcygeal GCTs were all reported on as part of larger studies on GCTs in multiple anatomical locations. Information on histopathology, staging and risk factors specifically affecting survival for sacrococcygeal GCTs were therefore limited. This prevents an accurate subclassification into Type I and II GCTs, although it is expected that the majority will be type I [47,48]

When detecting disease, AFP, beta-HCG and LDH are currently the only clinically suitable serum tumour markers. However, these markers are only elevated in ~60% of patients at the time of primary diagnosis [60,61]. Novel biomarkers with improved sensitivity and specificity would be of major clinical benefit. Several candidate molecular markers have been suggested, but they have yet to show clinical utility in the setting of liquid biopsies [62]. The miRNAs are a possible exception; they are crucial in early development and differentiation [63]. Embryonic miRNAs, specifically miR-371a-3p, have been investigated extensively and show great potential for the clinical management of testicular GCTs and, more sporadically, extra testicular GCTs [64,65,66,67,68]. Furthermore, these miRNAs might be explored as therapeutic targets in the future, including in the context of overcoming cisplatin resistance [69,70].

This review has some limitations. First, the different types of studies included made it difficult to pool data, and this precluded performing a meta-analysis of survival data. Second, the different chemotherapy regimens used internationally made it difficult to draw conclusions on treatment effect across anatomical localisations. Third, 10 out of our included studies were retrospective in nature, introducing the possibility of bias. This included the use of historical nomenclature as well, underrepresenting the developmental heterogeneity of GCTs that is currently recognised. Last, any systematic review relies on presenting formally published data, which leads to an inherent problem caused by publication bias. 

## 9. Materials and Methods

### 9.1. Search Strategy

The search was conducted following the Preferred Reporting Items for Systematic Review and Meta-Analysis (PRISMA) guidelines [71]. Studies investigating the treatment and outcome of malignant extracranial GCTs in children and young adults were identified via a literature search of PubMed, EMBASE and Medline. Studies, with a research population of patients aged 0–19 being considered as “paediatric”. Our overall search strategy included the MeSH heading “Neoplasms, Germ Cell and Embryonal” and the subheadings “Surgical Procedures, Operative”, “Radiotherapy”, “Chemotherapy, Adjuvant”, “Neoadjuvant Therapy” and “Treatment Outcome”, and it can be found in Appendix A. The resulting titles and abstracts were screened for relevance and in- and exclusion criteria, and the resulting articles were reviewed.

### 9.2. Study Selection

Inclusion criteria for this systematic review were articles focusing on the treatment and outcome of malignant extracranial GCTs in the paediatric population. As some large international protocols, e.g., MAKEI, include sex cord-stromal tumours in their treatment protocol, even though they are a separate entity from GCTs, we decided not to exclude articles from our review which include SCSTs. Exclusion criteria were case reports and case series including <20 patients to improve the level of consistency, letters to the editor, non-English literature, non-Western studies, non-human studies, studies on other cancers, studies on only mature and immature teratomas, studies on Type IV ovarian tumours, studies which did not include data on survival and patient age > 19 years old. If studies on children, young adults and adults included ≥20 children/young adults, they were included. Three authors (IM, CH, AS) performed manual reviews of the identified titles and abstracts. Studies including benign pathology were only included if survival data were reported separately for malignant and benign pathology, and the benign pathology data could be dismissed. Similarly, studies including a paediatric and adult population were only included if the results on the paediatric part of the included cases were reported separately. Full text review was performed by IM and CH. Any disagreement between the two authors was resolved by the third reviewer (AS).

### 9.3. Data Extraction and Analysis

The study characteristics and data points extracted from each study included anatomical tumour location, tumour stage, tumour histology, number of patients included, as well as patient age and gender. Different staging systems were used in the various studies. International Federation of Gynecology and Obstetrics (FIGO) stage 1A/B, Children’s Oncology Group (COG) stage I and postsurgical tumour nodes metastases (TNM) stage I were referred to as “stage I”. FIGO stage IV, COG stage IV and clinical or postoperative TNM stage IV were referred to as “stage IV”. The remaining patients were referred to as having “stage II/III” and considered as a single entity, because many large international trials report their data with stage II/III disease grouped together as an intermediate risk group. Information on treatment protocol, details of surgical and chemotherapy treatment and overall survival (OS) and event free survival (EFS) was also included. 

### 9.4. Statistical Analysis

A random effects model was employed for ovary, testis, mediastinal and sacrococcygeal localisation to estimate an overall effect. For each type of tumour, the number of cases at five years was approximated according to the number of patients and a given OS and EFS proportion. The inverse variance method, which gives more weight to larger studies, was used to pool EFS and OS for different studies. These are shown in the forest plot, together with the 95% Confidence Interval. The sizes of the square boxes on the forest plots are proportional to the total number of patients in each study. An overall test on heterogeneity between studies was performed for each meta-analysis (value I^2^). Heterogeneity was considered serious if the I^2^ value was >50% or *p* < 0.05 and very serious if I^2^ value was >80%. To estimate the between-study variance, which is represented as ‘tau’ in the forest plots, DerSimonian–Laird’s method was employed. All statistical analyses were performed in R software (R Foundation for Statistical Computing, Vienna, Austria).

## 10. Conclusions

In conclusion, the results of the current systematic review describe the heterogeneous nature of GCTs in different anatomical locations, impacting on stage at presentation, treatment modalities used and survival data. Despite this heterogeneity, subpopulations can be defined which have an inferior EFS and OS and where future research and more individualised treatment would help to improve survival.

## Figures and Tables

**Figure 1 cancers-13-03561-f001:**
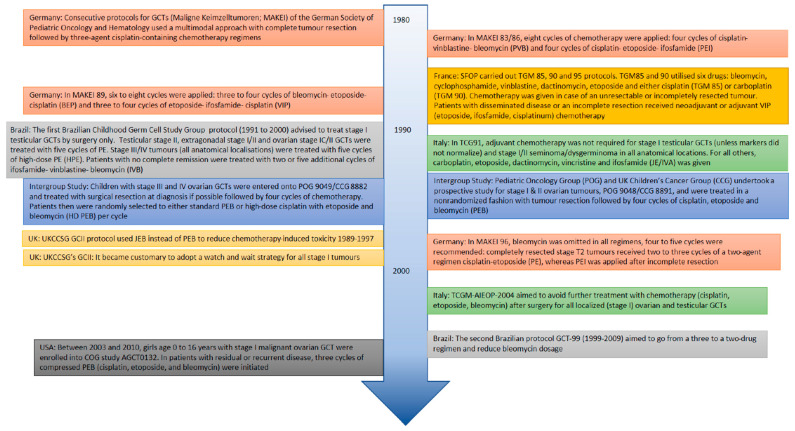
(inter)national protocols and intergroup studies over the past two decades.

**Figure 2 cancers-13-03561-f002:**
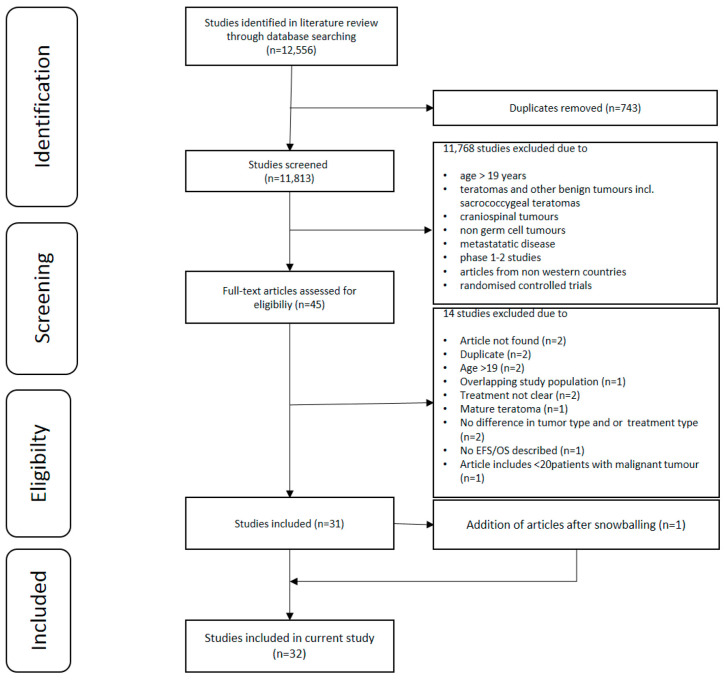
The selection process, based on the PRISMA schema.

**Figure 3 cancers-13-03561-f003:**
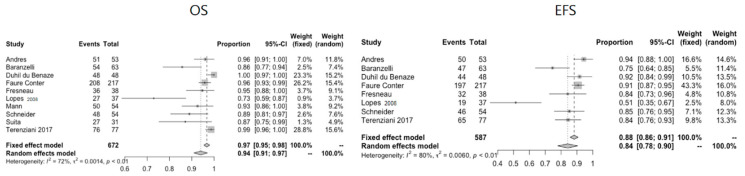
Pooled OS and EFS data for ovarian GCTs.

**Figure 4 cancers-13-03561-f004:**
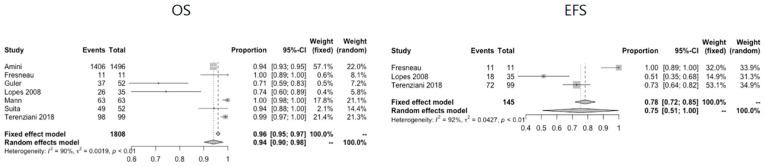
Pooled OS and EFS data for testicular GCTs.

**Figure 5 cancers-13-03561-f005:**
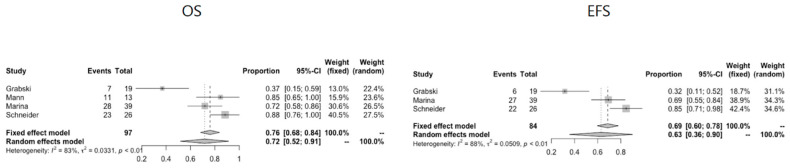
Pooled OS and EFS data for mediastinal GCTs.

**Figure 6 cancers-13-03561-f006:**
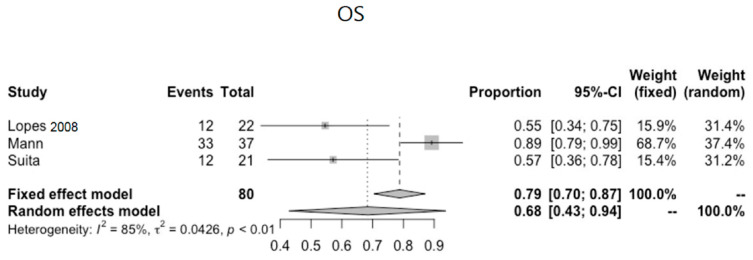
Pooled OS data for sacrococcygeal GCTs.

**Table 1 cancers-13-03561-t001:** Details on the studies of ovarian pathology.

Study	n	Location	Hist	Stage	Treatment Protocol/Chemotherapy Regimens	EFS (%)	OS (%)
Akyuz 2000	56	O	GCT	I (34%) II (27%) III (34%) IV (5%)	<’86 VAC ‘86–‘89 PVB ‘89–’97 BEP	57	68 overall. By stage: I = 82, II = 79, III = 50, IV = 33
Andres 2010	53	O	GCT	I (63%) II (4%) III (29%) IV (4%)	Unclear	94	97
Baranzelli 2000	63	O	GCT	I (2%) II (43%) III (49%) IV (6%)	TGM85 & TGM90	74	85
Billmire 2004	131	O	GCT	I (32%) II (12%) III (44%) IV (12%)	POG/CCG intergroup study BEP vs. HDBEP	No overall data. By stage: I = 95, II = 88, III = 97, IV = 87	No overall data. By stage: I = 95, II = 94, III = 97, IV = 93
Billmire 2014	25	O	GCT	Study on stage I disease only	COG AGCT0132	52	96
Lo Curto 2003	23	M	GCT	I (44%) II (4%) III (48%) IV (4%)	TCG91	81	88
Duhil du Benaze 2018	48	O	DYS	NM	TGM85 & TGM90 & TGM95	91	100
Faure Conter 2017	217	O	GCT	II/III (90%) IV (10%)	Protocols of 8 different trials COG/CCLG/SFCE	91 overall. By stage: II/III = 91, IV = 88	96 overall. By stage: II/III = 97, IV = 91
Frazier 2018	278	M	GCT	NM	PEB and JEB	PEB 90, JEB 85	NM
Fresneau 2015	38	OT	SCST	I (61%) III (39%)	TGM95	85	94
Lopes 2008	37	M	GCT	NM	’83–’86 VAB-6 ’87–’91 EPO-VAC ’91–’97 TCG91	51	73
Lopes 2009	45	M	GCT	NM	GCT91	NM	NM
Lopes 2016	206	M	GCT	NM	GCT99	91	92
Mann 2000	54	M	GCT	I (35%) II (17%) III (41%) IV (7%)	UK GC II	91	92
Rogers 2004	57	OT	GCT	I (72%) II (28%)	POG9048-CCG8891	93 overall. By stage: I = 95, II = 88	94 overall. By stage: I = 95, II = 94
Schneider 2003	54	O	SCST	I (88%) II (6%) III (6%)	MAKEI ‘83/86, ’89,’96	86 overall. By FIGO stage: Ia = 100, Ic = 76, II/III = 67	89
Shaikh 2017	124	M	GCT	NM	AGCT0132	3 cycles PEB = 88, 4 cycles PEB = 92	NM
Stern 2002	6	M	GCT	NM	JEB	NM	NM
Suita 2002	31	M	GCT	NM	Unclear	NM	87
Terenziani 2001	29	O	GCT	I (33%) II 15%) III (41%) IV (11%)	<‘80 VAC ’80-’87 PVB >’88 PEB	82	82
Terenziani 2017	77	O	GCT	I (35%) II (17%) III (42%) IV (6%)	TCGM-AIEOP2004	85 overall	99 overall
	**T = 1652**					By stage: I = 72, II–IV = 91 Dysgerminoma = 87	Stage I = 100, II–IV = 98 Dysgerminoma = 100

Hist = histology, O = ovary, OT = ovary testis, M = mixed anatomical locations, GCT = germ cell tumor also incl. sex cord stromal tumors and dysgerminomas, SCST = sex cord-stromal tumor only, DYS = dysgerminoma only, NM = not mentioned, VAC = vincristine actinomycin cyclophosphamide, PVB = cisplatin vinblastine bleomycin, BEP/PEB = bleomycin etoposide cisplatin, JEB = carboplatin etoposide bleomycin, VAB-6 = vinblastine bleomycin cisplatin cyclophosphamide actinomycin D doxorubicin, EPO-VAC = etoposide cisplatin vincristine, vincristine actinomycin cyclophosphamide.

**Table 2 cancers-13-03561-t002:** Details on the studies of testicular pathology.

Study	n	Location	Hist	Stage	Treatment Protocol/Chemotherapy Regimens	EFS (%)	OS (%)
Amini 2016	1496	T	GCT	I (57%) II/III (22%) IV (21%)	Unclear	NM	94
Cost 2014	59	T	GCT	I (48%) II (23%) III (29%)	Unclear	87 ped 60 adolescents	100 ped 85 adol.
Lo Curto 2003	36	M	GCT	I (73%) II (8%) III (11%) IV (8%)	TCG91	85	100
Frazier 2018	163	M	GCT	NM	PEB and JEB	PEB 85 JEB 100	NM
Fresneau 2015	11	OT	SCST	I (100%)	TGM95	100	100
Guler 2003	52	T	GCT	I (60%) II (15%) III (15%) IV (10%)	BEP, VAC, Vinbl&Bleo	NM	Overall 71. BEP 86, VAC 68, Vinbl&Bleo 64. Histology YST 74, Embryon.carc 64
Lopes 2008	35	M	GCT	NM	’83–’86 VAB-6 ’87–’91 EPO-VAC ’91–’97 TCG91	50	73
Lopes 2009	22	M	GCT	NM	GCT91	NM	NM
Lopes 2016	126	M	GCT	NM	GCT99	83	88
Mann 2000	63	M	GCT	I (72%) II (13%) IV (15%)	GC II (JEB)	100	100
Rescorla 2015	80	T	GCT	Study on stage I disease only	COG AGCT0132	74	100 overall. Age < 11 yrs 80 Age > 11 yrs 48. Histology pure YST 81, mixed GCT 81
Rogers 2004	17	OT	GCT	II (100%)	POG9048-CCG8891	100	100
Schlatter 2003	63	T	GCT	Study on stage I disease only	POG9048-CCG8891	79	100
Shaikh 2017	47	M	GCT	NM	COG AGCT0132	3 cycles PEB = 83 4 cycles PEB = 95	NM
Stern 2002	4	M	GCT	NM	JEB	NM	NM
Suita 2002	52	M	GCT	NM	Unclear	NM	93
Terenziani 2002	31	T	GCT	Study on stage I disease only	Surveillance (PEB/PVB for relapse/adolescents)	81	100
Terenziani 2018	99	T	GCT	I (59%) II (7%) III (14%) IV (20%)	TCGM-AIEOP2004	73 overall. By stage: I = 65, II = 100, III = 85, IV = 82	99 overall. By stage: I-III = 100, IV = 94
	**T = 2456**						

Hist = histology, T = testis, OT = ovary testis, M = mixed anatomical locations, GCT = germ cell tumor also incl sex cord stromal tumors and dysgerminomas, SCST = sex cord-stromal tumor only, NM = not mentioned, VAC = vincristine actinomycin cyclophosphamide, PVB = cisplatin vinblastine bleomycin, BEP/PEB = bleomycin etoposide cisplatin, JEB = carboplatin etoposide bleomycin, VAB-6 = vinblastine bleomycin cisplatin cyclophosphamide actinomycin D doxorubicin, EPO-VAC = etoposide cisplatin vincristine, vincristine actinomycin cyclophosphamide, YST = yolk sac tumor.

**Table 3 cancers-13-03561-t003:** Details on the studies of mediastinal pathology.

Study	N (M:F)	Location	Hist	Stage	Treatment Protocol/Chemotherapy Regimens	EFS (%)	OS (%)
Lo Curto 2003	2 (?)	M	GCT	NM	TCG91	NM	NM
Frazier 2018	69 (?)	M	GCT	NM	PEB and JEB	PEB 77 JEB 77	NM
Grabski 2016	19 (15:4)	Med	GCT	I (53%) IV (47%)	Unclear	29	39
Mann 2000	13 (?)	M	GCT	I (15%) II (23%) III (23%) IV (39%)	UK GC II	75	83
Marina 2006	39 (?)	M	GCT	NM	POG 9049/CCG 8882	69	71
Schneider 2000	26 (?)	Med	GCT	NM	MAKEI ‘83/86, ‘89, ‘96	83 overall. By completeness of surgery: incomplete surgical resection 42. Complete resection 94	87 overall. By completeness of surgery: incomplete surgical resection 42. Complete resection 100
	**T = 168**						

Hist = histology, Med = mediastinum, M = mixed anatomical locations, GCT = germ cell tumor also incl sex cord stromal tumors and dysgerminomas, NM = not mentioned, BEP/PEB = bleomycin etoposide cisplatin, JEB = carboplatin etoposide bleomycin.

**Table 4 cancers-13-03561-t004:** Details on the studies of sacrococcygeal pathology.

Study	n	Location	Hist	Stage	Treatment Protocol/Chemotherapy Regimens	EFS (%)	OS (%)
Lo Curto 2003	30	M	GCT	I (10%) II (3%) III (57%) IV (30%)	TCG91	59	70
Frazier 2018	169	M	GCT	NM	PEB and JEB	PEB88 JEB86	NM
Lopes 2008	22	M	GCT	NM	’83-’86 VAB-6 ’87-’91 EPO-VAC ’91-’97 TCG91	27	55
Lopes 2009	24	M	GCT	NM	GCT91	NM	NM
Lopes 2016	73	M	GCT	NM	GCT99	77	81
Mann 2000	37	M	GCT	I (5%) II (31%) III (17%) IV (47%)	UK GC II	87	88
Marina 2006	88	M	GCT	NM	POG9048-CCG8891	NM	NM
Stern 2002	9	M	GCT	NM	JEB	NM	NM
Suita 2002	21	M	GCT	NM	Unclear	NM	57
	**T = 473**						

Hist = histology, M = mixed anatomical locations, GCT = germ cell tumor also incl sex cord stromal tumors and dysgerminomas, NM = not mentioned, BEP/PEB = bleomycin etoposide cisplatin, JEB = carboplatin etoposide bleomycin, VAB-6 = vinblastine bleomycin cisplatin cyclophosphamide actinomycin D doxorubicin, EPO-VAC = etoposide cisplatin vincristine, vincristine actinomycin cyclophosphamide.

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
