# Peer review of "Treatment and Survival of Malignant Extracranial Germ Cell Tumours in the Paediatric Population: A Systematic Review and Meta-Analysis"

_cancers, 2021, doi:10.3390/cancers13143561_

Round 1

Reviewer 1 Report

The paper is a metaanalysis of studies published regarding extracranial germ cell tumours in the paediatric population.  It provides a nice summary of survival and treatment data on these uncommon neoplasms, analyzed by location, including ovarian, testis, mediastinum, sacrococcis.   The study attempts to shed some light on the diagnosis and management of these neoplasms, which have been classified differently both geographically and through the decades.

In my opinion, the manuscript could be greatly improved if several issues would be addressed by the authors:

  1. The authors analyzed overall survival and event free survival.  They do not specify what constitutes an event.  This is particularly important, as it is likely that the included studies analyzed survival differently.  
  2. The authors screened a total of 11813 studies.  After exclusion of 11768 studies, the number of studies assessed for eligibility dropped to 45.  Among the reasons for excluding papers was the focus on sacroccygeal location.  However, the authors do make an analysis of sacroccygeal location in the manuscript (n=473 cases from nine articles).  These were sacroccygeal tumors included in manuscripts that included mixed anatomical locations exclusively.  It is not clear why the authors chose to exclude those studies focused on sacroccygeal germ cell tumors, particularly if at the end they were going to do an analysis on them.  Potentially, sacrcoccygeal location-focused studies would have more reliable data than those which were not focused on this location.
  3. The authors appear to suggest that sex-cord stromal tumors are a subtype of germ cell tumor, and are analyzed as such for both ovarian and testicular tumors.  This is a nosologic mistake, as SCST are not a type of GCT.  Thus, these studies (or cases) should have been excluded from analysis.  Its inclusion generates confusion and may convey an erroneous concept to the uninformed reader.
  4. Given the retrospective nature of the study, the manuscript includes outdated terminology ("teratocarcinoma") and/or nosologic concepts (testicular mature teratomas), which unfortunately results in the perpetuation of the terminology confusion that has plagued these neoplasms throughout the years.  The authors do acknowledge this partially in the discussion ("The studies are a reflection of the world-wide approach of GCTS, which is varied and heterogeneous").  Since 2005, the Oosterhuis and Looijenga classification has attempted to introduce clarity and consistency in the classification of these heterogeneous groups of neoplasms across all anatomical sites.  This classification has been gaining traction in the recent years and is currently the basis for the current WHO classification of testicular neoplasms.  However, the authors only briefly mention this classification in the analysis of testicular tumors, missing an opportunity to explain why such discrepancy may be found in the clinical, pathologic, and molecular aspects of GCTs across different anatomic sites, or even within one anatomic site.  The paper would be greatly improved if the analysis were done with this classification in mind.

Reviewer 2 Report

This paper compiles the results of overall survival (OS) and event free survival (EFS) in pediatric patients with extracranial malignant germ cell tumors collected in the literature during the last 20 years. Analyzing, among other factors, how anatomical location, age or treatment affect OS and EFS.

Please review the following questions:

  1. Review all the acronyms used because many do not explain their meaning, for example:
  • Line 20: explain what GCTs means
  • Line 25 explain what AFS means
  • Line 61: explain what TGM is
  • Line 133: explain what AFS is
  • Line 137: LDH
  • Line 150: PVB
  • Line 202: YST
  1. Line 56-79: I recommend modifying and improving the wording. Some of the information is already included in the figure where it is better explained and the chronological order is better observed.
  2. Line 234: yrs replace by years or homogenize throughout the text if you want to use the abbreviation
  3. Expand the number of references, especially, it should improve the introduction and increase the number of references in that section
  4. Line 579: a format error, there is a nº 70 in the bibliography that does not correspond

Round 2

Reviewer 1 Report

The manuscript has been markedly improved by the authors revisions.  There persists one small inconsistency in regards to sacrococcygeal tumors.  The authors state in the "Study characteristics" section that the included cases "...predominantly reflect Type I GCTs."  However, when they describe the histopathology, they state that 62% had "mixed histology (teratoma and one or more malignant elements)malignant GCTs."   This last finding would suggest that in fact some if not most of the sacrococcygeal tumors included in the study corresponded to Type II GCTs.   This also conflicts with the statement in the discussion "This prevents an accurate subclassifications into Type I and II GCTs, although it is expected that the majority will be type I.  In fact, by excluding "benign teratomas" from the sacrococcygeal location in the included studies, the authors likely preferentially excluded Type I tumors, and thus is not surprising that a high proportion of included cases corresponded to Type II tumors.

Finally, the authors should mention that because of the exclusion of "benign teratomas" in the inclusion criteria, Type IV ovarian tumors (the most common GCT) was also preferentially excluded from the study.
